# Nanoparticle Additivation Effects on Laser Powder Bed Fusion of Metals and Polymers—A Theoretical Concept for an Inter-Laboratory Study Design All Along the Process Chain, Including Research Data Management

**DOI:** 10.3390/ma14174892

**Published:** 2021-08-27

**Authors:** Ihsan Murat Kusoglu, Florian Huber, Carlos Doñate-Buendía, Anna Rosa Ziefuss, Bilal Gökce, Jan T. Sehrt, Arno Kwade, Michael Schmidt, Stephan Barcikowski

**Affiliations:** 1Technical Chemistry I, Center for Nanointegration Duisburg-Essen (CENIDE), University of Duisburg Essen, 45141 Essen, Germany; ihsan.kusoglu@uni-due.de (I.M.K.); carlos.donate-buendia@uni-due.de (C.D.-B.); anna.ziefuss@uni-due.de (A.R.Z.); goekce@uni-wuppertal.de (B.G.); 2Institute of Photonic Technology, Friedrich-Alexander-University Erlangen-Nürnberg, 91052 Erlangen, Germany; Florian.Huber@lpt.uni-erlangen.de (F.H.); michael.schmidt@lpt.uni-erlangen.de (M.S.); 3Materials Science and Additive Manufacturing, School of Mechanical Engineering and Safety Engineering, University of Wuppertal, 42119 Wuppertal, Germany; 4Department of Hybrid Additive Manufacturing, Ruhr University of Bochum, 44801 Bochum, Germany; Jan.Sehrt@ruhr-uni-bochum.de; 5Institute for Particle Technology, Technical University of Braunschweig, 38104 Braunschweig, Germany; a.kwade@tu-braunschweig.de

**Keywords:** laser melting, laser sintering, 3D printing, AlSi10Mg, PA12, Round-Robin

## Abstract

In recent years, the application field of laser powder bed fusion of metals and polymers extends through an increasing variability of powder compositions in the market. New powder formulations such as nanoparticle (NP) additivated powder feedstocks are available today. Interestingly, they behave differently along with the entire laser powder bed fusion (PBF-LB) process chain, from flowability over absorbance and microstructure formation to processability and final part properties. Recent studies show that supporting NPs on metal and polymer powder feedstocks enhances processability, avoids crack formation, refines grain size, increases functionality, and improves as-built part properties. Although several inter-laboratory studies (ILSs) on metal and polymer PBF-LB exist, they mainly focus on mechanical properties and primarily ignore nano-additivated feedstocks or standardized assessment of powder feedstock properties. However, those studies must obtain reliable data to validate each property metric’s repeatability and reproducibility limits related to the PBF-LB process chain. We herein propose the design of a large-scale ILS to quantify the effect of nanoparticle additivation on powder characteristics, process behavior, microstructure, and part properties in PBF-LB. Besides the work and sample flow to organize the ILS, the test methods to measure the NP-additivated metal and polymer powder feedstock properties and resulting part properties are defined. A research data management (RDM) plan is designed to extract scientific results from the vast amount of material, process, and part data. The RDM focuses not only on the repeatability and reproducibility of a metric but also on the FAIR principle to include findable, accessible, interoperable, and reusable data/meta-data in additive manufacturing. The proposed ILS design gives access to principal component analysis (PCA) to compute the correlations between the material–process–microstructure–part properties.

## 1. Introduction

Powder bed fusion using laser beam (PBF-LB) [1] is a sub-class production technique of additive manufacturing (AM). Highly complex 3D structures and tailor-made designs can be produced by melting metal and sintering polymer powder feedstocks. Scientific studies in AM showed an exponential growth within the last decade, and studies on PBF-LB of metal and polymer powder feedstocks followed the growth trend [2,3]. Besides the scientific contributions, progress in the industrialization of PBF-LB can be seen in the increasing number of machine suppliers [4,5,6,7]. In addition, PBF-LB plays an essential role as a critical technology in the context of industry 4.0 as parts and products can be tracked. Therefore, the production chain is flexible and can be automated, leading to communication between machines, parts, and products [8] and understanding the metallurgical phenomenon of the PBF-LB process together with mechanistic models and machine learning opening a path to control the process by machines further and increase the reproducibility of the as-built parts with the repeatable microstructure and part properties [9,10]. However, current industrialization efforts for the PBF-LB technology need standardization in testing powder feedstocks, processability, and as-built part properties. Increasing repeatability and reproducibility in PBF-LB highly depends on these standardization activities. The ISO committee (ISO/TC 261) and ASTM committee (F42) work in cooperation to develop and publish new standards on PBF-LB and AM technologies [11,12].

Recent PBF-LB progress focuses on increasing metal and polymer powder compositions’ variability, developing processability, and enhancing part properties [13,14,15,16,17,18,19,20,21,22,23,24]. In order to overcome material-related limitations and expand the application fields of PBF-LB technology, recent studies address the nano-additivation of metal and polymer powder feedstocks [2,3,25,26,27,28,29,30,31,32,33,34,35,36,37,38,39,40,41,42]. Several material types of nanoparticles (NPs) can be supported on the surface of the metal and polymer powder feedstocks [2,3,25], leading to enhanced material properties, such as flowability, laser absorbance, and thermal conductivity. Furthermore, NPs are known to enhance the processability [26,27], microstructural properties by avoiding grain-growth [43,44,45,46], and as-built part properties [3,25,30,33,36,42], as well as broaden the process window [28,37,38,39,40,41].

Several reviews [2,3,47,48,49] have been published to understand the process–structure–property relationships in the PBF-LB process but a well-designed inter laboratory study (ILS) to test nano-additivated commercial powder feedstocks can further consolidate industrialization, standardization, robustness, and fundamental understanding along the entire process chain of PBF-LB. ILSs allow the statistical evaluation of measurable metrics’ repeatability and reproducibility linked to material, process, or as-built part properties. Participants’ repeatability and reproducibility of each measurable metric are valuable seed data to develop and standardize test procedures in the PBF-LB process. 

Several ILSs are designed and organized for PBF-LB of metal and polymer powder feedstocks [50,51,52,53,54,55,56,57,58,59,60]. Previous ILSs on PBF-LB focused on quantifying the variability in an as-produced and used state of metal [50,51] and polymer [52,53] powder feedstock properties without focusing on processability and as-built part properties. They included measurements of powder properties such as powder size distribution (PSD), shape, chemical composition, crystallographic phases, material rheology, flowability, and thermal behavior [50,51,52,53]. However, they showed that the statistical evaluation of powder properties is affected by the experimental setup, calibration in measurements, and powder modification during handling. Previous ILSs on PBF-LB also statistically evaluated the deviations in process parameter sets and as-built part properties of commercially available metal and polymer powder feedstocks [50,51,52,53]. However, these studies did not include a statement on powder properties that affect processability, microstructural formation, and as-built part property. Additionally, it could be demonstrated that the variability of as-build part properties between-participant is higher than within-participant [54,55,56,57,58,59,60], resulting in a strict recommendation to follow a manufacturing plan. 

To overcome these limitations, the proposed design herein includes the measurement of as-produced and used powder properties and evaluates process parameter sets and as-built part properties, including microstructural formation with at least 20 PBF-LB process participants (10 polymers, 10 metals). Besides testing as-produced powder polymer and metal properties along the entire process chain, nano-additivated feedstocks are included in the ILS design. The design is instructive for non-profit organizations, industry, and national or international standardization organizations conducting ILS to test new metal or polymer powder feedstocks developed for the PBF-LB process. The design includes an organizational workflow, sample flow, and data flow that evaluate the entire process chain based on decentralized PBF-LB processing, but centralized testing. In addition, this design focuses on the research data management that further evaluates the generated data by PCA. Reusable data opens a path to obtain relevant seed data that data analysts can further analyze for additive manufacturing and powder material design. The proposed ILS herein includes evaluating 27 powder property metrics, 12 process property metrics, 26 part property metrics for three metals (unmodified and two nano-additivated), and three polymers (unmodified and two nano-additivated) powder feedstocks along the entire PBF-LB process chain.

## 2. Design of Interlaboratory Study

The ILS organization has to be well-structured to trace work and material flows through processing and testing. Moylan et al. [61] briefly defined an organizational structure, which they recommend following in an ILS. Additionally, Maier et al. [62] confirmed that such an ILS’s primary limitation is an organization that needs particular, high scientific and management knowledge. A general organizational workflow of ILS specific to PBF-LB is given in Figure 1.

The advisory board has to be selected from research institutes and the AM industry. Important requirements are long-term experience in powder feedstock production, PBF-LB of metal and polymer powder feedstocks, material characterization techniques, and conducting ILS. However, it is good to include a specialist working on developing international standards for AM technology. The study director should have excellent managing skills and closely communicate with the study coordinator. The latter has a critical position in organizing, tracking, and evaluating the workflows’ progress with scientific research data management experience. The organization includes coordinating workflows in ILS to statistically evaluate the repeatability and reproducibility of NPs additivated powder feedstock and as-built part properties. 

Note that previous ILS [50,51,52,53,54,55,56,57,58,59] started introducing commercially available powder feedstocks followed by measuring one or two properties of the as-built parts for statistical evaluation of the measured metrics’ repeatability and reproducibility between different machine users. However, the ILS proposed herein includes testing NPs additivated metal and polymer powder feedstocks; hence, the ILS design needs to have additional workflows. These workflows have to test both unmodified and additivated powder feedstocks to measure NPs’ effect on powder material properties. Further, it includes testing NPs’ effect on microstructural formations and mechanical properties after the PBF-LB process, as shown in Figure 2. In that way, traceability along the entire process chain is enabled and the ILS will be completed in six steps, with each step being discussed in the following sections.

In each sub-section, the amount of acquired datasets delivered by the respective ILS activity is estimated. These datasets will be tagged with the user, time, SOP, and may mathematically consist of one-dimensional (e.g., flowability value), pairs (e.g., meting-crystallization temperature), or multidimensional data (e.g., spectra or histograms). The term “data” will be used through the manuscript and counted as one for one tagged dataset, independent of its dimensionality.

### 2.1. Additivation of NPs on As-Produced Metal and Polymer Powder Feedstocks

Before starting additivation of commercially available powder feedstocks, highly reproducible NP-additivated powder feedstocks are required to minimize material-dependant deviations in the ILS statistical evaluation. While as-produced metal and polymer powder feedstocks show good reproducibility and can be commercially produced on a ton scale, this is not the case for powder feedstocks in the development phase. In addition, up-scaling limitations in NP production and feasibility of additivation techniques can limit producing high amounts of NP additivated powder feedstocks for the PBF-LB process. The proposed ILS herein will require less than 700 kg feedstock material (187.5 kg polymer, 502.5 kg metal) for 20 PBF-LB processing participants (10 PBF-LB/P and 10 PBF-LB). There is a base amount of powder needed (1 kg polymer, 3 kg metal) for property assessment activity (see Figure 3) before the powders can be sent out to processing partners; after that, the required powder mass scales linearly with the number of participants that run PBF-LB (details on the build chamber volumes are given in chapter 2.1.2.). For example, 10 powder recipients each for PBF-LB/P and PBF-LB/M would require a total of 30 and 90 kg of polymer (3 different compositions) and metal powder (3 different compositions), and 20 each for PBF-LB/P and PBF-LB/M requires 247.5 and 592.5 kg of polymer and metal powder for the execution of the whole ILS.

As mentioned above, the inclusion of nanoparticles in the powder feedstock represents a new challenge that has been ignored to date in ILSs and takes a central point in this ILS. The production of nano-additivated powder feedstocks requires the supporting of NPs on the surface of commercially available metal and polymer powder feedstocks. Hence, knowing the NP properties before and after supporting is crucial and needs a proper analysis beforehand. The main requirement is a high reproducibility of the whole synthesis process, including the NP synthesis route and the additivation process. As shown in Figure 3a, the surface coverage and the distribution of NPs on the micro-powder’s surface can vary depending on the NP load and surface additivation technique. Additionally, NPs may be present as a dry powder [41] or in a colloidal solution [28,30]. These variations may affect powder feedstock material properties relevant for PBF-LB.

Lüddecke et al. [40] studied the powder characteristics of several metal powders coated with different NP material types, sizes, and supporting amounts (<1 vol%) in fluidized bed coatings. They showed a comparison of the flowability behavior of nano-additivated metal powders by measuring bulk density, unconfined yield strength using ring shear tests, and dynamic angle of repose within a rotating drum. They found that the flowability of powder feedstocks can be improved by the optimum NP size, NP material type, and loading amounts. Hupfeld et al. [28,29,32] presented a simplified scaling diagram for NP supporting on micro powders, link in NP particle size, surface coverage (surf%), and vol% as a prerequisite to creating very well defined, highly dispersed submonolayer NP coatings on PA12 and TPU, including the comparison between wet and dry coating. Pannitz et al. [39] showed that coating SiC NPs or few-layer graphene (FLG) NPs on stainless steel powder feedstocks decreases the laser reflectivity of the powder feedstock and increases the thermal conductivity of the material during processing, leading to a rapid heat dissipation into the already solidified underlying layers during the melting of next powder layer. These different studies show the importance of measuring the physical and chemical properties of NPs before producing the additivated metal and polymer powder feedstocks to probe correlations between NP property, supported powder property, and the behavior along the process chain during ILS. Hence, before and after supporting NPs on the ILS´s powders, complementary measurement techniques are recommended to determine the physical and chemical properties of NPs, summarized in Figure 3.

It has been shown that the chemical composition of NPs does not change during the supporting process, but may change during processing, where the NPs see high temperatures and may undergo aggregation in the melt during processing [33,36,63]. Different material types of NPs, such as metals, oxides, non-oxides, and compounds, can be used as additives in powder feedstock formulations [3,25]. The crystalline phases of NPs should be identified by X-ray diffraction (XRD), a common material characterization technique. X-ray fluorescence (XRF) is the recommended, non-destructive characterization technique to determine the elemental composition of NPs and trace the chemical formations and elemental compositions in as-built parts after the PBF-LB process [64]. Therefore, within the ILS, XRD and XRF methodologies are recommended to determine the NP’s chemical structure before starting the additivation process.

The size distribution of NPs is a vital factor in tracing the reproducibility of NPs. Further, NP size distribution must be known to calculate the amount of NPs needed for an intended surface coverage level on the micro-powders [29]. It is known that the NP´s size influences the powder feedstock flowability [40,41] as attractive forces between micro-powders are reduced by roughening the surface with NPs [65,66]. However, a statistical investigation is still missing in the literature. Following Gaertner et al. [41], the feedstock’s flowability behavior will be enhanced by optimum surface coverage level and optimum NPs size. As discussed in references [67,68], transmission electron microscope (TEM), dynamic light scattering (DLS), and analytical disc centrifugation (ADC) characterization techniques are recommended methods within an ILS to measure dry and colloidal NP size distributions.

A widespread approach to measure dry NP size distribution is TEM, but careful sample preparation is needed, and depending on the NP size, hundreds to thousands of particles must be measured for statistical relevance [67]. Moreover, TEM may create drying artifacts and lack good size statistics. Instead, widely employed techniques to measure the hydrodynamic size distribution of NPs are DLS and ADC. While DLS lacks in determining multimodal size distributions, ADC is capable of measuring polydisperse colloids. Detailed information on the comparison between TEM, DLS, and ADC can be found elsewhere [67,68], including the comparison of five analytical methods for particle diameter differentiation and bimodality identification [69]. Please note that the particle size can alter during the supporting process, mainly forced by agglomeration-induced effects [41]. Therefore, we advise determining both supported and colloidal NP size distribution.

#### Additivation Process

The surface additivation of the metal and polymer powders can be performed by dry mixing or wet coating and subsequent or parallel drying [28,33,37,40,41]. The choice of coatingtechnique highly depend on the availability of NPs. Dry powder mixing techniques such as three-dimensional free-fall shaker [41] and rotary mixing drum [37] can be used. Then, the mixing parameters such as duration and rotating speeds need optimization for homogeneous dispersion of NPs on the surface of powder feedstocks. Gärtner et al. [41] studied the effect of dry-coating SiO2 NPs (<1 vol%) on the flowability behavior of several particle size fractions of CoCrFeNi alloy powder. They showed that short dry mixing durations avoided NP agglomeration on the metal powder surface and increased the powder flowability. Homogeneous NP distribution on the metal and polymer powder surface [28,33,37,40,41] will result in reproducible powder feedstocks and as-built part properties. 

Supporting colloidal NPs requires a wet coating technique followed by additional drying [27]. Fluidized bed drying is a suitable method where the powders can be wet mixed and dried in the same unit with a relatively short processing time, facilitating minimum cross-contamination [26,40]. Another wet mixing technique while wet mixing the materials is dielectrophoretic deposition [28,29,30]. It depends on the stirring medium’s pH, the concentration of the materials, isoelectric points of NPs, and powder feedstocks in the medium [67]. Key to the dielectrophoretic deposition is the particle’s surface charge density, which promotes this process. For a progressive wet surface additivation, NPs and micro-powders should be oppositely charged. For this purpose, the isoelectric points of both materials should be measured under different pH values before starting the additivation process. Further information on this process can be found elsewhere [19,32,42,67]. The resulting mixture must be sieved and dried before using the PBF-LB process [28,29,30,31,32,33]. This ILS is designed to test six different powder composition configurations as unmodified and 2 different NP additivated metal as well as unmodified, and 2 different NP additivated polymer powder feedstocks. 

The total amount of powder production is a crucial point for a complete run in ILS. It depends critically on the height of the build job, the build area of each participant’s machine, and the number of participants in PBF-LB processing. Participating PBF-LB machines can have different specifications and build areas. The build job area and height must be minimized to avoid excessive powder consumption in the ILS. The build job height has to be defined before starting the ILS, and is defined by the longest dimension of the Z-direction specimen. In detail, a build area of 35 cm × 35 cm and height of 10 cm will have a volume of 12.25 cm^3^ per build job. Polymer powder such as polyamide-12 (PA12) with a theoretical density of almost 1 g/cm^3^ and a relative powder bed density of 50% requires 6.25 kg of powder feedstocks per build job. Likewise, metal powder such as AlSi10Mg (2.68 g/cm^3^) having a relative powder density of 50% will need 16.75 kg of powder feedstocks per build job. An ILS design testing three PA12 powder compositions with 10 participants requires 187.5 kg of powder feedstocks, and three AlSi10Mg powder compositions with 10 participants need 502.5 kg of powder feedstocks. Some PBF-LB participants may have a smaller build area, e.g., 25 cm × 25 cm, in their machine, which will reduce the total amount of powder requirement to run the ILS. 

Further, it should be noted that the height of the build volume for polymer PBF-LB will include an additional base layer for thermal decoupling from the metal platform with top layers on top of the finished parts. In contrast, the height of the build volume for metal PBF-LB will include an additional support structure. Therefore, some PBF-LB machines can have a smaller build area than the planned build job, not allowing participation in the designed ILS. In this case, splitting build jobs can enable participation. In addition to the participant’s process failures, each machine’s recoating mechanism must be considered as well in calculating the total amount of powder required for the ILS. The volume of the powder conveyor can be different from machine to machine. However, additional powder for recoating the first layers will be needed, and overdosing in powder recoating is required to spread a good powder layer during processing. Supplying a double amount of powder feedstock to each participant per build job will further accelerate ILS progress (e.g., 200 kg of PA12 and 500 kg of AlSi10Mg powders in total for 10 participants and 3 powder variants processing a build volume of 6250 cm^3^ per build job). After delivery by the ILS study coordinator, participants must store the delivered powder feedstocks containers according to the provided SOP conditions. 

### 2.2. Powder Quality of Unmodified and NP-Additivated Powder Feedstocks 

Since feedstocks’ material properties can differ from company to company, all unmodified metal or polymer powder feedstocks should be supplied from the same metal and polymer powder producer. The powder chemical composition, size distribution, shape, flowability, thermal behavior, and laser interaction of metal or polymer powder feedstocks are the most reported material properties affecting the laser processability microstructure, porosity, and build parts’ mechanical properties [2,3]. Depending on the physicochemical properties and volumetric loadings of NPs, additivated powder feedstock material properties will be differentiated from unmodified powder feedstock material properties. In addition to the material properties given above, surface occupation density (surface coverage %) and interparticle distance between NPs covering the surface must be measured to differentiate volume loading from surface dispersion effects. Within the proposed ILS, each participant’s feedstock batch must be measured before starting the PBF-LB process, which will later be correlated with statistical deviations in each participant’s as-built part properties and microstructural analysis. The material properties recommended to be analyzed within this ILS design are discussed in the following, and the recommended techniques to measure those material properties (Figure 4) are given in the following sub-sections. The number of data generated for each property metric is shown in Figure 4 and evaluated in the material data generation (Section 2.6). Therefore, in the second step of the ILS (Figure 2), both unmodified and additivated powder feedstocks’ material properties as chemical composition, shape, size distribution, flowability, thermal behavior, laser interaction, moisture content, surface occupation density of NP, and interparticle distance of NP on the micro powder surface will be measured.

A corresponding characterization of the unmodified metal and polymer powder feedstocks must reference the additivated powder’s data in this context. The repeatability of measurements and reusability of generated data (see Section 2.6) needs standardization in sample preparation and testing conditions, which will be described in Section 2.5.

#### 2.2.1. Chemical Composition 

The chemical composition of an unmodified metal or polymer powder feedstock directly affects the laser interaction, process window, microstructural formations, and as-built part properties [2,3,35]. These properties can be further affected by NPs’ chemical composition, supported amount, and surface coverage levels [29,30,31,32,33,38,39]. In this ILS, the chemical composition of powder feedstock is recommended to be measured by XRF and ICP-OES. Measuring the weight % of the matrix element (i.e., wt% Al in AlSi10Mg alloy), first major element (i.e., wt% Si in AlSi10Mg alloy), and NP matrix element (the highest wt% metallic element of NP phase) will generate a total of 160 items of data for metals and 40 items of data for polymers (Figure 4). Note that XRF cannot measure the polymer’s elemental composition but the amount of NPs in the polymer matrix. Alternatively to XRF, ICP-OES can be used within the ILS, but the polymer digestion method must be carefully executed for ICP-OES analysis of composites to avoid matrix cross-effect [70]. Measuring chemical composition of feedstocks by XRF and ICP-OES will generate 160 data points for metals and 40 data points for polymers (Figure 4).

#### 2.2.2. Powder Shape and Powder Size Distribution 

Powder bed density is directly linked to powder shape and size distribution during the PBF-LB process [45]. Additionally, different powder size fractions using the same process parameters influence the part quality [2,3,71,72,73]. Hence, in the ILS, the shape of the powder feedstocks has to be measured by dynamic image analysis. Measuring the sphericity of powder feedstocks will generate a total of 30 data points for metals and 30 data points for polymers (Figure 4).

The particle size distribution (D10, D50, and D90) will be measured by statistically relevant laser diffraction and dynamic image analysis [18]. Thus, a total of 180 data points for both metals and polymers will be generated as material data in ILS (Figure 4). 

#### 2.2.3. Flowability 

PBF-LB is a layer-by-layer powder processing technique, and in each layer, the recoater spreads powder feedstocks on the building platform. During spreading, powder feedstocks must have excellent flowability to increase powder bed density and facilitate recoating a homogeneous and defect-free powder layer [74]. Since there is no international standard to measure PBF-LB powder feedstocks’ flowability, a combination of several techniques such as density measurement (Hausner ratio, HR), ring shear test, and the rotating drum has to be used to measure the unmodified and additivated powder feedstocks’ flowability. More details on the mentioned techniques can be found elsewhere [37,40,41,60,61,75].

In short, measuring the HR is a simple technique defined by the ratio between tap density and bulk density of powders [52]. Values smaller than 1.25 describe free-flowing characteristics [53,75]. Besides measuring the bulk density of powders, HR test conditions do not mimic the dynamic powder deposition conditions of PBF-LB. For this purpose, a ring shear test and the rotating drum test have to be used within the ILS to understand the dynamic behavior of powders. Note that the ring shear test measures the so-called ffc ratio (consolidation stress/unconfined yield strength) of quasi static (low shear rates) and the powders and classify those from non-flowing to free-flowing (with values > 10 [40]), and the rotating drum test is another technique to determine the flowability and cohesion of powders within dynamic movement [76] in which the dynamic angle of repose (dAoR) and cohesive index (CI) is linked to the flowability of powders. The measurements at different rotation speeds are appropriate to determine the recoating speed limitations during the PBF-LB process [37,40]. Avalanche angles smaller than 45° describe excellent to good flowability [41], and a cohesive index greater than 0 describes good flowability [40]. In this ILS, the flowability measurements by the three methods will generate 150 data points for metals and 150 for polymers (Figure 4).

#### 2.2.4. Thermal Behavior 

Depending on the material types, powder feedstocks have different melting and crystallization temperatures during the heating and cooling stages of PBF-LB. These temperatures can be shifted by introducing NPs [27,28,38]. Metal powders entirely melt during the PBF-LB process. The as-built metal parts can result in over 99.5% of theoretical density after processing [2]; however, this is often not the case for polymer powder feedstocks. By increasing the laser energy densities, polymer feedstocks will partially melt to the point where sintering and elastoplastic flow mechanisms between powders occur. Kusoglu et al. [3] showed that over 99% of densification in the as-built part has rarely been achieved for most thermoplastics. The processing window, the difference between the onset melting and crystallization temperature, determines the required build chamber temperature to process polymer powder feedstocks. The processing window of nano-additivated powders depends on the NP material type and volume loadings, affecting melting–crystallization temperatures [28,59]. Differential scanning calorimetry (DSC) is a well-known approach to measure the melting and crystallization temperatures of unmodified and additivated powder feedstocks to predict the processing window of PBF-LB/P. Additionally, powder bed temperatures can be set according to the process window to delay rapid crystallization during the PBF-LB process. The DSC heating rate is too low compared to the PBF-LB process, but the cooling rate can be well represented by DSC between 0.2 and 20 K/min in PBF-LB/P. However, there are different cooling rates throughout the powder bed volume, which is difficult to mimic for DSC. A new approach of differential fast scanning calorimetry (DFSC) with very rapid heating and cooling rates can represent the PBF-LB process conditions better, to understand the rapid heating and cooling behavior of metal and polymer powder feedstocks [27,28,62]. Derived from DSC and DFCS, a total of 120 data points for metals and 120 data points for polymers will be generated by measuring the melting and crystallization temperature of the metal and polymer powders (Figure 4). 

#### 2.2.5. Laser Reflectivity 

PBF-LB is a process where powders are heated up to melting temperatures of the powder material. The laser absorbance or reflectance of powders is known to influence the required PBF process parameters and melt pool dynamics. Laser interaction with NPs, unmodified metal powders, unmodified polymer powders, and additivated powders will differ depending on powder crystalline structure and surface chemistry (affecting the NP´s absorbance at the laser wavelength). As seen in the inset of Figure 3a, by increasing the NP load, the surface coverage differentiates. Additionally, depending on the degree of surface coverage, the laser reflectivity of metal and polymer powder feedstocks will be different after NPs additivation [40]. Diffuse reflectance infrared Fourier transform spectroscopy (DRIFTS) measures the reflectivity (scattering) of powder feedstocks for a wide spectral range depending. Ideally, the DRIFTS is equipped with a heating unit to heat the powder feedstocks to the PBF bed’s temperatures [76]. The reflectivity measurements should be performed under inert gas atmosphere (Ar for metals and nitrogen for polymers) to mimic the PBF-LB process conditions. The reflectivity at a wavenumber of 943 cm^−1^ corresponds to wavelength 10.6 µm. This characteristic laser wavelength of carbondioxide (CO_2_) lasers is mainly employed in the PBF-LB of polymer powder feedstocks. The reflectivity at a wavenumber of 9398 cm^−1^ corresponds to a wavelength of 1064 nm (emission of the Nd:YAG lasers), primarily used in PBF-LB of metal powder feedstocks. DRIFT measurements will generate a total of 60 data points for metals and 60data points for polymers at room temperature and powder bed temperature (Figure 4).

#### 2.2.6. Moisture Content 

The moisture content is another property that adversely affects flowability, processability, and pore formation during PBF-LB. Therefore, if a wet additivation is used, the powder feedstocks must be dried until weight loss is completed in a condition avoiding degradation or oxidation of powders. Note that moisture absorption from the atmosphere can lead to gas pores forming within the as-built part during metal or polymer powder feedstocks processing. Therefore, it is recommended to dry the powders under vacuum followed by storage under inert gas. The moisture content of the metal and polymer powders will be measured by thermo gravimetric analysis (TGA), and a total of 30 data points for the metal and 30 data points for the polymer will be generated (Figure 4).

#### 2.2.7. Surface Coverage of Micro-Powders by NPs and Interparticle Distance between NPs

It is known that powder size distribution, flowability, powder bed density, the powder’s laser interaction properties, and the additivation technique are influenced by the surface coverage value of NPs on the micro powders [39,40,41]. This surface coverage is highly dependent on NP size distribution, NP load, and additivation technique [28]. SEM has to be employed in the ILS to measure the surface occupation density of NPs and inter-particle distance between NPs on the surface of metal and polymer powder feedstocks. It is a time-consuming image analysis technique, and tens to hundreds of additivated micro-powder particles must be imaged and measured for statistical relevance [76]. A total of 60 data points for the metal and 60 data points for the polymer will be generated by measuring surface coverage % and the average distance between NPs on the surface of micro powders (Figure 4).

Several techniques to measure unmodified and NPs additivated metal and polymer powder feedstocks’ material properties were described in Section 2.2. During ILS, it is strongly recommended to measure each property by the same operator using the same equipment in a central laboratory (Figure 1). The study coordinator has to deliver standard operational procedures (SOPs) for each measurement technique that central laboratories must follow. Available international standards for conducting related tests must be embedded in these SOPs (see Section 2.6).

### 2.3. PBF-LB Process

After material properties of unmodified and additivated metal and polymer powder feedstocks are tested and embedded in the RDM, the powder feedstocks are sent out to the processing participants in the third step of ILS (Figure 2). It should include the manufacturing plan SOPs, process control documents (PCDs), technical drawings, standard triangle language (STL) files, and, if possible, building plates for metal parts. Every participant must use the central STL and additive manufacturing file format, which defines the whole geometry, specimen placement, and build job support structures. For example, every test specimen must be labeled, e.g., with numbers to indicate its position in the build job. Since tens of specimens will be built in each build job, all samples with built-in unique numbers will be formatted in STL. In doing so, participants will not have to mark each built specimen after PBF-LB, facilitating traceability. The position in the building envelope and parts orientated in different directions will be traceable even during post-processing. STL files have to be sent to participants at least one month before the start of ILS. CAD drawings will be converted to STL files by a central entity. Later, STL files will be distributed to each process participant to ensure that everyone uses the duplicate files. It is known that CAD to .stl conversion can be challenging in very complex parts. Our ILS design mainly addresses the powder material qualification and processability question rather than creating complex 3D parts. Hence, printing is based on simple geometries (i.e., cubes, bars, and cylinders) where the CAD to .stl conversion is not challenging. Each process participant will use their build processor for data preparation that fits their PBF-LB machine. Various machines and software versions of participants can have problems introducing the STL of the build job [61]. In this case, the study coordinator must be informed immediately to decide further participation in the ILS. Machine users must carefully follow the PBF-LB process SOPs, including allowed parameter corridors, e.g., laser power, scanning speed, hatch distance, and powder layer thickness. 

Several strategies can be followed in processing powder feedstocks, while it is highly dependent on each participant’s PBF-LB machine type, specifications, and experiences. Previous ILSs on processing unmodified powder feedstocks showed that a strict manufacturing plan with a narrow corridor range of process parameters (shown in Figure 5) could be supported if each participant’s machine type is identical, which is often not the case. Otherwise, between-participant deviations in as-built part property can be high [54,61]. Another strategy can be, selecting experienced institutions in processing unmodified metal or polymer powder type of ILS. In this case, proposed for the present ILS, each participant can set their best practice processing parameters within the corridor range given in the manufacturing plan [57]. As a limitation to best practice process parameters of participants, achieving a certain level of relative density and/or UTS in unmodified as-built parts is requested from the participants, e.g., relative density (RD) > 99%, ultimate tensile strength (UTS) > 400 MPa for AlSi10Mg [2], and RD > 90%, UTS > 48 MPa for PA12 [3]. As a result of the second strategy, participants will individually set their best-effort process parameters shown in Figure 5. Process parameters will include laser power, scanning speed, hatch spacing, powder layer thickness, laser beam diameter, powder bed temperature, contour parameters, powder recoater speed, chamber atmosphere, and laser wavelength. Each participant must report the process parameter sets shown in Figure 5 into PCDs and send them to the study coordinator after processing. PCDs will be used as a meta-data source during ILS data evaluation. Each participant must process the metal build job on the building plates supplied by the study coordinator and send it to the coordination office without separating as-built parts. A total of 360 data points both for metals and polymers will be generated as the process data (Figure 5).

### 2.4. Powder Quality of Used Powder Feedstocks

At the end of the PBF-LB process, each participant must remove an aliquot of the unprocessed powder feedstocks, ideally 2.5 kg for AlSi10Mg and 1 kg for PA12 powders. These used powder aliquots must be stored in the containers without sieving and sent back to the ILS coordination office with the build jobs. As a fourth step of ILS (Figure 2), each participant’s used unmodified and NP-additivated metal and polymer powder feedstock properties will be analyzed with the same test methods stated in the second step (see Section 2.2) to draw conclusions on the powder´s reusability and property changes. PBF-LB of metal powders may cause spatters and agglomerates, e.g., direct ejections from the molten pool, the dragging of particles by the gas stream over the process zone. These spattered powders might be sintered and have been reported to deviate from the chemical composition of the as-produced feedstock powders and degradation due to heat, oxidation, and change in the powders surface morphology have been observed during PBF-LB [51]. Used powder quality is another interest in ILS to determine the material properties of used metal and polymer powder feedstocks for reusability and statistical interlaboratory dependence in the PBF-LB process. Therefore, it is highly recommended to test the material properties of used powder feedstocks after sieving. The sieve must match the measured biggest powder size of the as-produced metal and polymer powders, e.g., a sieve size > DV, 99 of metal and polymer powder feedstocks.

### 2.5. As-Built Parts

All metal and polymer build job parts must be delivered to the coordination office before proceeding to the ILS’s next step. The coordination office will organize preparing specimens and tests at the central laboratories in the fifth step (Figure 2). As-built polymer parts will be built on the powder layers without support structures, and after the dimensional accuracy of each part is measured, polymer specimens will be sent to the assigned central laboratories for testing.

Compared to PBF-LB/P, building a metal part needs a support structure in the PBF-LB/M process. These support structures are being manufactured along with the parts on the building plate. As-built parts must be separated with a cutting machine such as electrical discharge machining (EDM). 

Metal alloy PBF-LB parts are usually heat-treated to reduce residual stresses in the as-built parts or rearrange microstructural formations affecting the mechanical properties of as-built parts. If heat-treatment is of interest in ILS, as-built parts must be heat-treated before measuring part properties as given in the next section of the ILS plan. The effect of heat-treatment conditions on mechanical properties of the AlSi10Mg alloy is given in ASTM F3318-18 standard [77]. Depending on the heat-treatment conditions, microstructural formations in the metal matrix, residual stress in the as-built parts, tensile strength, and tensile elongation is improved compared to the as-built state [78]. As mentioned in the introduction, NPs may have a strong impact on microstructural formations during processing that affect the mechanical properties of as-built parts, requiring a different heat-treatment condition. If heat treatment is included in the ILS, unmodified and NP-additivated specimens should be heat-treated in a single central lab using the same furnace and heat treatment conditions. The coordination office must provide related SOPs to central labs for metal part separation and heat-treatment. However, this ILS is designed to test directly as-built conditions in order to minimize the required powder amount and specimens to be characterized (that would double in case of heat treatment is included, as understanding of the material response would requires analysis of both the heat-treated and as-built specimen). In this case, after cutting, removing support structures, and measuring the dimensional accuracy of the specimens, the as-built parts are sent to assigned central laboratories to test as-built part properties. 

Following sample preparation, testing the effect of NPs on the as-built part properties of each participant’s build job will be carried out in the fifth step (Figure 2). Both powder properties and process parameter sets affect the microstructural formations, porosity, and mechanical properties of as-built metal and polymer parts [2,3,13,17,18,30,34,36]. Therefore, the effect of NPs on microstructural and mechanical properties has to be investigated with the test methods shown in Figure 6. Composition, relative density, porosity, NP imaging, and tensile properties will be measured for both metals and polymers. Matrix grain size and crystal orientation will be measured separately for metal parts; molecular weight and crystallinity will be separately measured for polymer parts to assess the parts’ microstructural formations.

#### 2.5.1. Microstructural Formations

NPs are known to rearrange microstructural formation in both metal and polymer parts [2,3,25,26,28,29,30,33,36]. Since metal and polymer are different material types, investigation techniques to understand the effect of NPs on microstructural formations need different analytic techniques. The grain size of metal parts is one of the measurable properties to understand the effect of NPs on microstructural formation. NPs can act as nucleation sites during the cooling of the melt pool, which fines the grain size in the microstructure, and a decrease in grain size can be linked to an increase in tensile strength by the Hall–Petch equation [36]. SEM will observe metallographically prepared surfaces/cross-sections of the as-built parts. Measurements have to be carried out both from the vertical and horizontal building directions of the sample according to the ASTM E112 [79] standard. 

To investigate the NP effects on the crystallographic orientation during the melt pool’s solidification, electron backscattered diffraction (EBSD) measurements in the as-built metal parts [36] are implemented in the ILS. At least 3 replicas should be used for EBSD analysis that generates 540 data points for metals (Figure 6). The effect of NPs on the polymer parts’ microstructure will be determined by measuring the parts’ molecular weight and crystallinity. The molecular weight of the as-built polymer parts will be measured by gel permeation chromatography (GPC). GPC analysis showed that depending on the process parameter sets, e.g., different build chamber temperatures affect the polymer’s average molecular weight, and a change in chain length alters the crystallization temperature of the polymer [80]. The crystallinity degree of as-built polymers will be analyzed by polarized light microscopy and DSC [80,81]. Crystallinity measurements will be performed on 3 replicas and generate 270 data points for polymers (Figure 6).

#### 2.5.2. Chemical Composition 

A defined number and type of NPs are supported on the feedstocks, and the chemical composition of additivated metal and polymer powder feedstocks are measured in the second step of ILS. However, it is unknown if all additivated NPs will be transferred to the as-built metal and polymer parts during processing, as losses during powder handling or laser melting cannot be excluded. Furthermore, some metal alloy elements can be vaporized during laser melting that affects the solidification microstructure and part quality [43]. As discussed above (Section 2.2.1), XRF [27,33,40] and/or ICP-OES [36] will measure the as-built polymer and metal parts’ chemical composition. Measurements by XRF have to be carried out on the polished surface of metal and polymer cubes. 3 replicas will be measured by non-destructive XRF, and the same replicas will be sent to relevant central laboratory to conduct ICP-OES measurements. Measurements by XRF and ICP-OES will generate 480 data points for metals and 120 data points for polymers (Figure 6). The transferability of the amount of additivated NPs along the process chain will be determined by comparing the chemical composition (nanoparticle vol%) of unmodified and processed, as well as unmodified and additivated powders with as-built parts. 

#### 2.5.3. Relative Density and Pore Size Distribution

Porosity in as-built metal and polymer parts significantly impacts the density, mechanical, and functional properties [57,58,59,60]. The microstructural formations and porosity distributions of the as-built parts can be affected by the dissolution degree, lattice misfit degree, high-temperature chemical stability, wetting degree, interface reactions, and laser absorptivity of NPs during PBF-LB processing [36,82,83,84,85,86]. Further, the parts’ relative density and porosity content can affect as-built parts’ mechanical and functional properties [2,3]. After the PBF-LB process, as-built parts can contain pores, and the effect of NPs on the final density and pore size distribution must be determined. Volume µ-CT and cross-sectional optical microscope (OM) measurements must be performed to measure the relative density, volumetric porosity content, and volumetric pore size distributions of as-built polymer and metal parts [58,59,60]. 3 replicas will be measured for each material. Relative density and pore size distribution will generate 720 data points both for metals and polymers (Figure 6).

#### 2.5.4. NP Imaging

Measuring the distribution and sizes of NPs in the metal or polymer matrix is essential for evaluating the as-built part properties. Depending on the material properties of NPs, different reaction mechanisms can occur with metal melt and polymer melt that affect the microstructural formation in the as-built parts [36,38]. For example, NPs can dissolve in metal melt and, during the cooling stage of PBF, can form nano- or micro- precipitates with a different crystalline phase than starting NPs structure [33,36]. On the other hand, undissolved NPs in the melt pool can rearrange in the metal matrix during the cooling stage of the PBF-LB process. As illustrated in Figure 7, laser power or energy density variations can affect the melt pool’s undissolved the NPs rearrangement in the melt pool. Marangoni convection significantly impacts NP rearrangement in the metal melt pool [63,86,87]. Depending on the laser power or energy density, surface tension gradients between melt and solid drive a fluid flow in the melt pool. Increasing laser power or energy density will increase the Marangoni convection and melt pool depth during processing. In addition, there can be other factors in melt pool dynamics: evaporation in the melt pool, recoil pressure, and inverted Marangoni flow direction due to inhomogeneous melt pool chemistry [36,63,75,78,86,87].

Conducted ILSs showed that deviations in the as-built part property are high between participants rather than within-participant properties [54,61]. As stated in Section 2.3, participants can set different laser energy densities within a given range to process the powders. Depending on the process parameter sets, the melt pool dynamics can vary between participants, and NP rearrangement in the as-built parts can vary, affecting microstructural formation. Therefore, the final status of NPs in each participants’ as-built part must be investigated. Thin cross-sections will be prepared from 3 exemplary replicas (randomly selected each for polymer and metal) by microtome for polymer parts and focused ion beam (FIB) for metal parts before conducting TEM measurement. Thin cross-sections of both vertical and horizontal directions will be investigated by TEM to measure the NPs sizes in the polymer or metal matrix [28,36]. TEM measurements will generate 120 data points both for metals and polymers (Figure 6). Since there is no standard test method for this investigation, an SOP must be prepared to follow the same experimental and analysis procedure in cross-section TEM investigations. 

#### 2.5.5. Static Mechanical Properties

Previous ILSs showed that the mechanical properties of the unmodified parts could vary between participants [54,56,57]. The tensile test is the most common method that can quantify the effect of NPs on as-built parts and can be used both for polymer and metals. Depending on the building direction within the build job volume, parts can exhibit different tensile behavior [55,57]. It is highly recommended to build test specimens in three directions: vertical, horizontal, and diagonal to the building direction. The dimensions, geometry, and testing conditions have a high impact on the tensile test results. There is no international standard yet precisely defining tensile testing of as-built metal and polymer parts processed by PBF-LB. As stated previously, minimizing the tensile specimen’s length will decrease the building height of the build job, so the total amount of powder requirement in the ILS run will be reduced. By conducting tensile tests, the effect of NPs on the ultimate tensile strength (UTS), yield strength (YS), elastic modulus (EM), and tensile elongation (TE) of metal and polymer parts can be measured. Therefore, 6 replicas will be built in vertical, diagonal, and horizontal build directions for each polymer and metal build jobs. It is recommended to test metal tensile specimens with a height/length of 60 mm (Form B, DIN 50125:2016-12) and polymer tensile specimens with a height/length of 49 mm (Form C, DIN 50125:2016-12) within this ILS. Tensile specimens will be tested according to DIN EN ISO 6892-1 standard for metals [88] and ISO 527-2:2012 standard for molding and extrusion plastics [89]. Tensile tests will generate a total of 2160 data points both for metals and polymers (Figure 6).

Overall, the effect of NPs on volumetric relative density, microstructural formation, and mechanical properties of the as-built polymer and metal parts (Figure 6) will be measured for each participant’s build job in the fifth step of ILS (Figure 2).

### 2.6. Research Data Management (RDM) of ILS

Since this is the first time a proposed PBF-LB ILS evaluates a comprehensive set of powder material, process, microstructure, and part properties, there will be a vast data source at the end of the ILS. On the one hand, this rich data pool is an ideal basis for statistical assessments and correlation function extraction along the full process chain, from the powder to the part. On the other hand, solid statements and robust data processing, in particular correlation analysis, is only possible if data and meta-data management is strictly embedded into the ILS concept. Therefore, it is essential to set up an RDM plan beforehand and follow FAIR principles [90]. In this manner, transparent, reproducible, and reusable data can be provided by the available research process with all documented components. Here, data registration plays a critical role in the success of the ILS RDM. Each central laboratory will document the sampling techniques and test conditions while measuring each property metric, collecting metadata, and generating datasets of related property metrics to improve data registration. All datasets with their corresponding documentation and SOPs will be stored in a central repository created by the coordination office. A long-term accessible repository will give open access to validated data for future data-mining interests in the field of AM. Data reporting should be performed according to ASTM F2971-13 [91]. In addition, ASTM is developing a new standard (ASTM WK73978 [92]) to comprise actions for data registration which will draw a route map for high-quality data registration under FAIR principles in the field of AM. Then, all the data can be further analyzed by data miners or linked to the further ILSs to advance discovery in material selection and design of AM.

Furthermore, this traceable data forming and structuring will set the base for functional correlations in PBF-LB via PCA. An exemplary data flow is prepared to show the complexity of ILS design and the importance of research data management to further evaluate the generated data before starting the ILS. As shown in Figure 8, the dataflow matrix systemically shows the data generation for the selected process variables in the ILS design. The dataflow matrix shown in Figure 8 is designed for ten polymer and ten metal PBF-LB process participants testing unmodified and two different NPs additivated metal and polymer powder feedstocks.

In the sixth step of the ILS, the ILS design’s data generation is classified into three sections along the process chain: material, process, and parts. The material data of as-produced (second step, Section 2.2) and used (fourth step, Section 2.4) unmodified and NPs additivated powder feedstocks per participant have to be evaluated to compare the properties of as-produced and used powder feedstocks in the sixth step (Figure 2). As shown in Figure 5 and Figure 8, in total of 9 variables (chemical composition, powder size distribution, powder shape, flowability, thermal behavior, laser interaction, moisture content, surface occupation density, and interparticle distance between the NPs on the surface of micro powders) will be evaluated by measuring a total of 52 property metrics. A total of 790 data points for metals and 750 data points for polymers will be evaluated to compare the as-produced and used powder properties. The material data generated from as-produced, unmodified, and NP- additivated powder feedstocks will be correlated with the generated process and part data by PCA (Figure 8).

A total of 20 PCDs filled by participants will be sent back to the study coordinator after PBF-LB process. By doing so, each participant’s 12 PBF-LB process properties (Figure 5) will be extracted to generate a total of 360 data points both for metals and polymers (Figure 8), and further will be correlated with the microstructure and part properties to determine the effect of 20 participants’ process parameter variation on the unmodified and nanoadditivated metal and polymer as-built part properties. 

The part data generated from as-built part properties of unmodified and NP-additivated powder feedstocks per participant are statistically evaluated in the sixth step (Figure 2). As shown in Figure 6 and Figure 8, in total 5 variables (composition, density, pore size distribution, average NP size within the as-built parts, the tensile properties of both metals and polymers), an additional 3 variables (grain size, crystallographic texture, and misorientation) for metals, and 2 variables (molecular weight, crystallinity) for polymers will be evaluated by measuring a total of 49 property metrics. 

As shown in Figure 8, a total of 65 property metrics in PBF-LB of unmodified and NP-additivated metal and polymer powder feedstocks will be statistically evaluated in ILS. Following the FAIR principle in data generation, subsequent PCA allows a correlation between material–process–part properties to understand the PBF-LB process’s complexity. For example, Kusoglu et al. [2,3] conducted a PCA of a small data subset by extracting the reported powder material, process, and part properties data from the most-cited 100 SCI-Expanded articles published within the last decade for PBF-LB of Al alloys and polymers. A data matrix was generated in that study from 139 Al powder compositions [2] and 257 polymer powder compositions [3] with the corresponding material, process, and as-built part properties. Unfortunately, only 33 Al powder compositions and eight polymer powder compositions could be successfully processed by PCA because of missing information parts in most published work. As a result, only one powder property, six process properties, and three as-built properties were successfully correlated with these limited datasets. These studies [2,3] showed the importance of reporting the same properties, using international standards in test methodologies, building the same test specimen geometries, and following FAIR principles. 

The statistical evaluation of ILS data will focus on calculating the repeatability and reproducibility of each property metric for between-participant and within-participant test results. ASTM E691-20 [93] described how to conduct an ILS to determine a test method’s precision. Luping and Schouenborg [94] showed the equations to calculate the repeatability/reproducibility of the test results. These equations should be used in the proposed ILS´s statistical evaluation. Both repeatability and reproducibility conditions strictly rely on using the same test methods to measure identical material within a short interval.

Additionally, repeatability conditions benefit from the same operator’s tests at the same laboratory using the same equipment, which is embedded in the ILS design by the concept of central testing laboratories. With SOPs, reproducibility conditions depend less on the tests at different laboratories by different operators using different equipment. Repeatability and reproducibility limits will be allowed to a probability of 95% between two test results and can be obtained by multiplying sr or sR by a factor of 2.8 as a rule of thumb. Since the reproducibility depends on conducting the tests under the same experimental conditions in different laboratories, reproducibility can be calculated for the same build jobs processed in different PBF-LB machines. Finally, the consistency of the test results is assessed by graphical techniques as Mandel’s k, which is a within-participant consistency statistic, and Mandel’s h, which is a between-participant consistency statistic [93,94].

A total of 68 property metrics will be evaluated for unmodified and additivated metal and polymer powder feedstocks in the suggested ILS design. Internationally accepted standards should be used to define each property’s testing conditions. By doing so, significant variations in the test results can be minimized. As a result, each property metric’s repeatability and reproducibility in original data can be increased. For this purpose, it is strongly recommended to prepare SOPs for each test method or property. Published and developing standards considered in the ILS design for the SOPs are given in Table 1.

Besides testing the properties of additivated metal and polymer powders, it is recommended to prepare additional SOP for NP characterization. Available standards as ASTM F1877-16 (characterization of the morphology, number, size, and size distribution of particles) [110], ASTM E2834-12(2018) (particle size distribution of nanomaterials in suspension) [111], ASTM E3247-20 (measuring the size of nanoparticles in aqueous media using DLS) [112], ISO 17200:2020 (characteristics and measurements of nanoparticles in powder form) [113], ISO/TR 14187:2020 (surface chemical analysis of nanostructured materials) [114] have to be used to prepare SOPs to test the properties of NPs shown in Figure 4. 

## 3. Recommended Implementation Procedure

Due to the complexity of the design testing new powder feedstocks in PBF-LB, it is not recommended to start directly with the full width of the ILS before pre-testing the properties of unmodified and NP additivated powder feedstocks, defining the range of various PBF-LB process parameters, and testing the sample flow and workflow of the intensive structural characterization in central labs. The basic workflow should be tested in a minimalized pre-test with 2 processing partners. The following control on the workflows, data generation, and cycle duration should be traced in pilot run (2–4 processing partners, central labs) before running the full-run ILS design (200 processing partners, central labs, research data management).

In this manner, ILS must be completed in three parts, (i) pre-test, (ii) pilot run, and (iii) full run. The purpose of each part is given in Figure 9.

Besides planning workflows, the study director and study coordinator must trace the logistics’ progress and timing already during the pilot run. Controlling logistics between the workflows will help draw flexible time planning for the full run and avoid ILS logistic-based delays. 

## 4. Conclusions

The implementation of interlaboratory studies in additive manufacturing is indispensable for further developing new materials for 3D printing methods such as PBF-LB. Here, a well-designed ILS can consolidate progress in industrialization, standardization, robustness, and fundamental understanding along the entire process chain of PBF-LB. Further, it is essential to accurately evaluate measurable property metrics and calculate property’s repeatability and reproducibility between ILS participants.

Although several ILS exist, no study describes the complete process chain based on different feedstock materials. We herein designed such a large-scale interdisciplinary ILS, ensuring a comprehensive analytical characterization along the process chain. In existing ILSs, a focus often lies on (e.g., part property) repeatability to manufacture specific parts (to see, e.g., machine-dependences) or to see inter-laboratory effects of general LPBF manufacturing of a material class (e.g., steel powder from varying sources). Such designs do not make sure that a process chain parameter can later be firmly attributed to another, not yet known one, via PCA. Additionally, the strong focus on a) powder feedstock properties and b) RDM is unique. To provide a more concrete picture of the general theoretical concept and ILS design, and provide exemplary numbers to the proposed RDM concept, the presented exemplary design statistically evaluates the entire process chain by measuring 27 powder property metrics, 12 process property metrics, and 26 part property metrics. Of course, an extension or reduction in the number of varied material properties is easily possible based on the presented general ILS concept. Further, the effect of nano-additives on the commercially available metal and polymer powder feedstocks and the repeatability of the modifications by NP can be measured for a better understanding of the PBF-LB process chain. Hence, this ILS design may be a blueprint for future ILS on LPBF, in particular in those ILS where powder feedstock effects shall be understood, and/or robust statistical analysis and PCA data processing is intended, where a good RDM design before ILS execution is key. The implementation of nano-additivated powders is not required, but is recommended, as it allows an expansion of the existing feedstock materials towards enhanced processability and shows further advantages in terms of avoiding crack formation, rearranging grain structure, and adding functionality to the as-built parts.

The ILS includes the experimental procedure and an RDM which is essential in a study of this scale. It enables an in-depth PCA based on FAIR principles in research data management. We highly recommend dividing the ILS into three stages—pre-test, pilot run, and full run—to successfully establish the latter for pure and nano-additivated feedstock materials. While the pre-test and pilot run of this ILS mainly focus on understanding the effect of powder properties, processability, and as-built properties during the PBF-LB process, the full run is required to prove its feasibility.

Testing new powder feedstocks within an ILS performed with a statistically relevant number of participants will guarantee reproducibility for both metal as well as polymer processing. Further, the ILS will set the data for determining correlations along the entire process chain and help to evaluate the available characterization techniques used for quality control of feedstocks and as-built parts in the field of additive manufacturing. 

## Figures and Tables

**Figure 1 materials-14-04892-f001:**
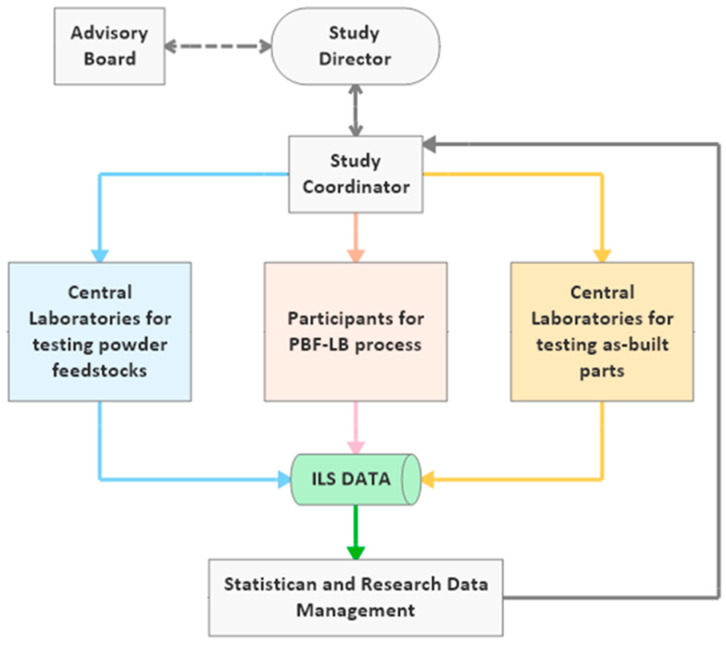
The organizational workflow of inter-laboratory study (ILS) design for powder bed fusion using laser beam, including decentralized processing and centralized testing.

**Figure 2 materials-14-04892-f002:**
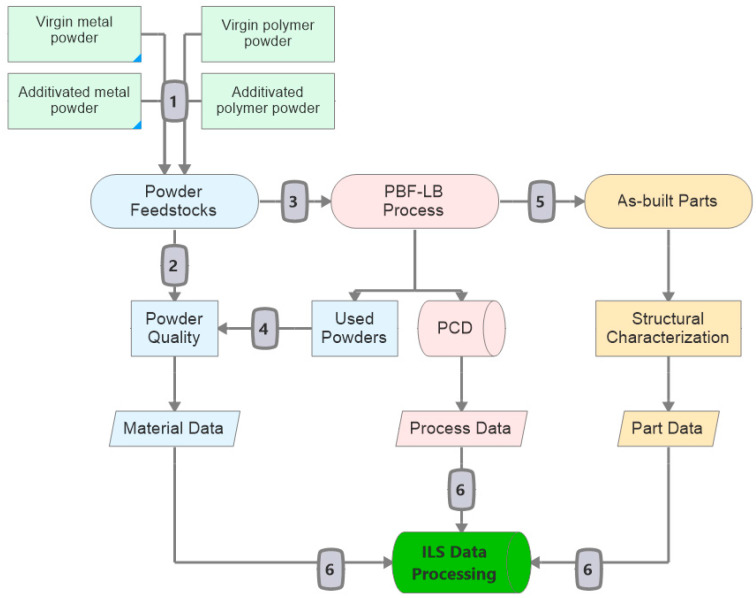
Workflow of inter-laboratory study (ILS) for powder bed fusion using laser beam (PBF-LB) of unmodified and NP-additivated metal and polymer powder feedstocks. The workflow consists of six steps: (1) nano-additivated powder production, (2) powder quality of unmodified and nanoparticle additivated powders, (3) PBF-LB of unmodified and additivated powders, (4) powder quality of used powders after the PBF-LB process, (5) testing as-built parts, (6) powder, process, part data generation, and ILS data processing. Topics in blue, pink, and orange represent the workflows to obtain the powder, process, and part data, respectively.

**Figure 3 materials-14-04892-f003:**
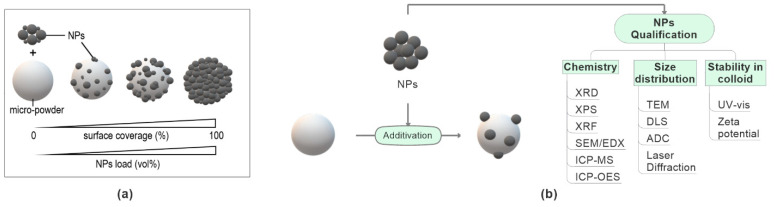
(**a**) Illustration of the surface coverage by micro powders with increasing nanoparticle loading (vol%) and (**b**) the most relevant nanoparticle properties for the additivation process and recommended nanoparticles characterization techniques.

**Figure 4 materials-14-04892-f004:**
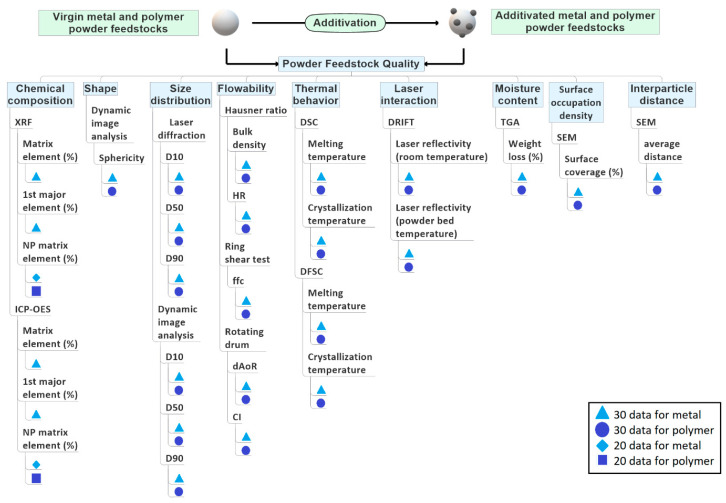
The material properties of powder feedstocks affected by nanoparticles recommended test methods to measure each material property for 30 metal powder batches and 30 polymer powder batches, and the number of data generated for each property metric. Explanation for abbreviations are as follow: X-ray fluorescence (XRF); inductively coupled plasma-optical emission spectroscopy (ICP-OES); the portion of particles with diameters smaller than this value is 10% (D10); the median diameter (D50); the portion of particles with diameters below this value is 90% (D90); hausner ratio (HR); consolidation stress/unconfined yield strength (ffc); dynamic angle of repose (dAoR); cohesive index (CI); differential scanning calorimetry (DSC), differential fast scanning calorimetry (DFSC); diffuse reflectance infrared fourier transform (DRIFT); thermogravimetric analysis (TGA); scanning electron microscope (SEM).

**Figure 5 materials-14-04892-f005:**
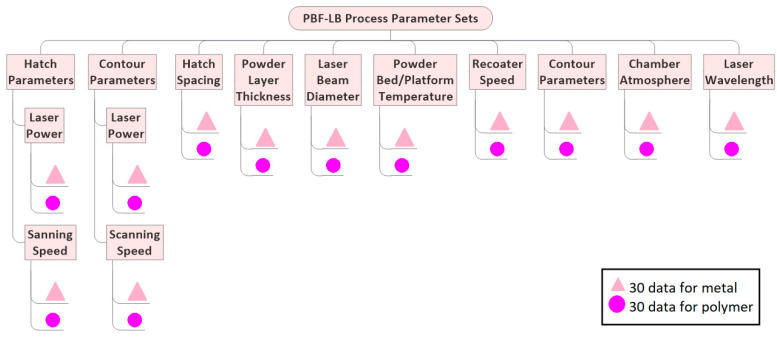
Participant-dependent PBF-LB process parameter sets and number of data generated for each property.

**Figure 6 materials-14-04892-f006:**
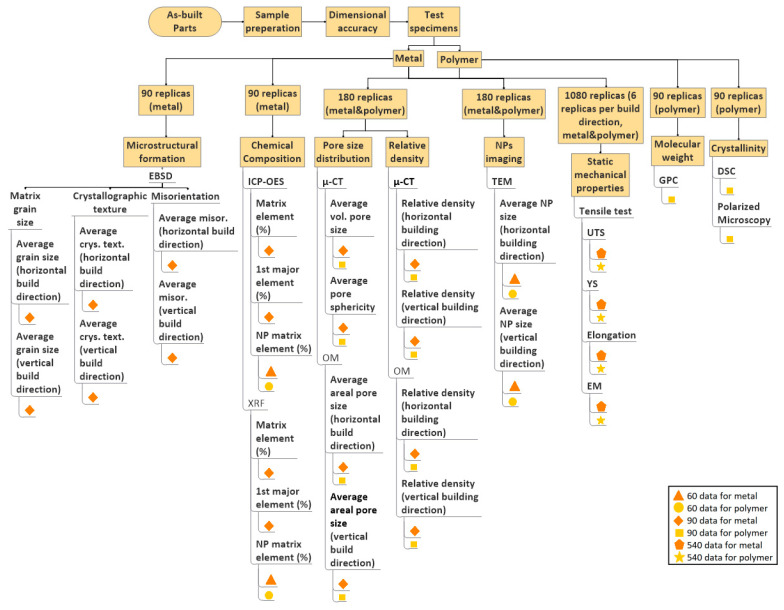
As-built part property analysis with related measurement techniques for nanoparticle-additivated metal and polymer parts, number of replicas for each property measurements per build job, and number of data generated for each property metric. Electron backscattered diffraction (EBSD), inductively coupled plasma-optical emission spectroscopy (ICP-OES); X-ray fluorescence (XRF); micro-computer tomography (μ-CT); optical microscope (OM); transmission electron microscope (TEM); ultimate tensile strength (UTS); yield strength (YS); elastic modulus (EM); gel permeation chromatography (GPC); differential scanning calorimetry (DSC).

**Figure 7 materials-14-04892-f007:**
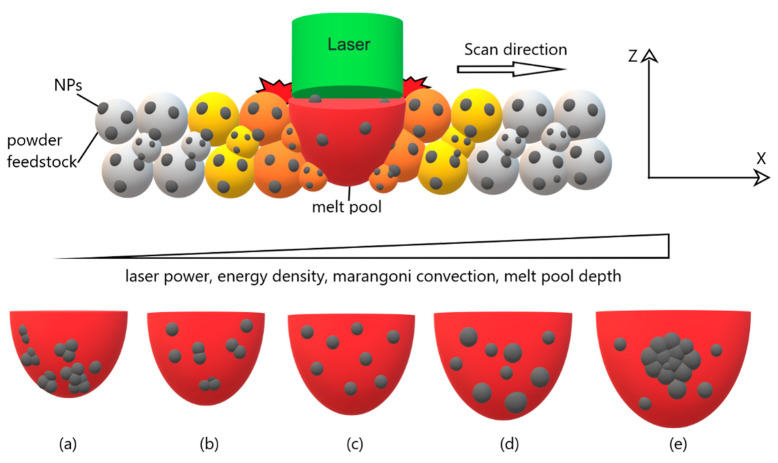
Illustration of nanoparticle distribution in the melt pool during laser powder bed fusion process. Depending on the process conditions, within the melt pool volume, the undissolved nanoparticles can be (**a**) poorly mixed and aggregated at the bottom, (**b**) loosely accumulated with improved distribution, (**c**) homogeneously distributed without accumulation, (**d**) well distributed but coarsened, and (**e**) highly aggregated to (sub)micrometer scale.

**Figure 8 materials-14-04892-f008:**
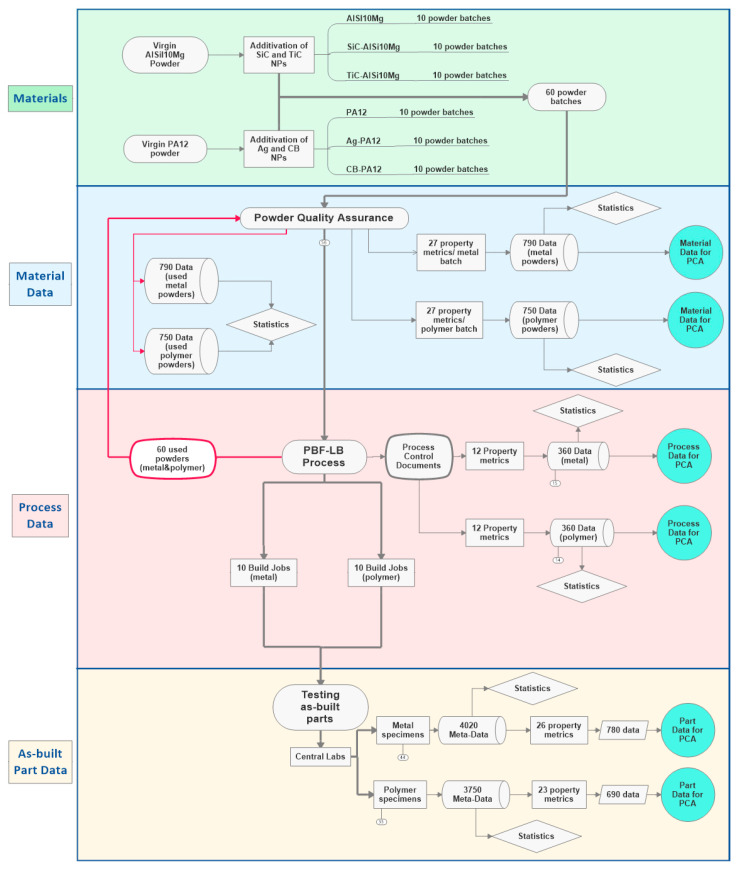
Material, process, part data flow chart of inter-laboratory study, and the entire process chain. Numbers are based on testing 1 unmodified and 2 nanoparticle-additivated powders, each for metal and polymer powder with 10 processing participants (each for polymer and metal) plus central lab participants.

**Figure 9 materials-14-04892-f009:**
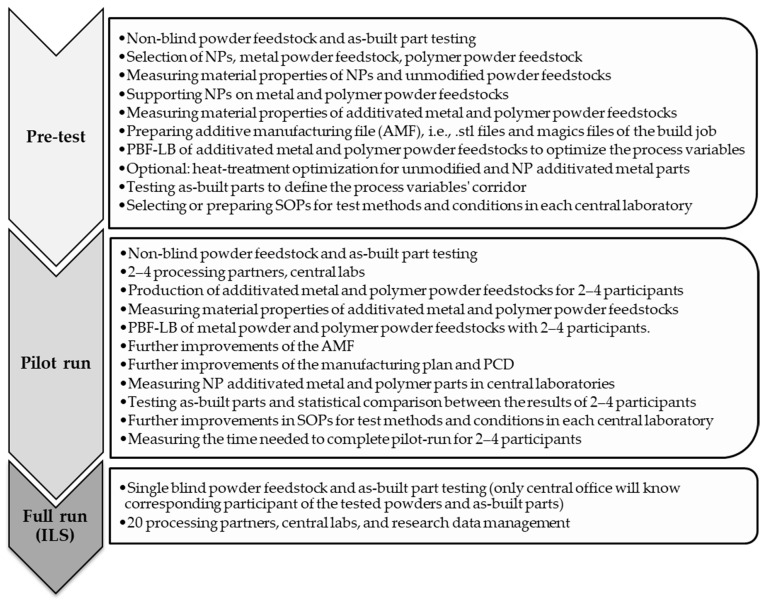
Design of the stepwise inter-laboratory study implementation via execution of pre-test, pilot run, and full ILS run.

**Table 1 materials-14-04892-t001:** Published and developing international standards on the characterization of laser powder bed fusion of metal and polymer powder feedstocks and as-built parts, as basis for the standard operational procedures of the inter-laboratory study.

Material	International Standards	Content
**Metal** **Powder**	ISO/ASTM 52907-19 [95]	Technical specification of as-produced and used feedstocks
	ASTM F3049-14 [96]	Test methods for powder size, morphology, chemistry, flowability, and density
	ASTM B527-20 [97]	Test method for tap density
	ASTM B822-20 [98]	Particle size distribution by light scattering
	ISO 13320:2020 [99]	Particle size distribution by laser diffraction
	WK66030 [100]	Quality assessment guidelines for powder reusability
	WK55610 [101]	Powder dynamic flow properties
	WK74905 [102]	Particle shape analysis to identify agglomerates/satellites
**Polymer Powder**	ISO 13320:2020 [99]	Particle size distribution by laser diffraction
	WK55610 [101]	Powder dynamic flow properties
**As-built Metal** **Part**	ASTM F3122-14 [103]	Evaluating mechanical properties
	ASTM E572-13 [104]	Measuring chemical composition by wide wavelength XRF
	ASTM E8/E8M-16ae1 [105]	Tensile test
	WK49229 [106]	Orientation and location-dependent mechanical properties
	ASTM E3166-20 [107]	Non-destructive examination of as-built parts
**As-built Polymer Part**	ISO 527-1:2012 [108]	General principles of tensile test
	WK66029 [109]	Tensile test

## Data Availability

All the data is available within the manuscript.

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
