# Peer review of "Nanoparticle Additivation Effects on Laser Powder Bed Fusion of Metals and Polymers—A Theoretical Concept for an Inter-Laboratory Study Design All Along the Process Chain, Including Research Data Management"

_materials, 2021, doi:10.3390/ma14174892_

Round 1

Reviewer 1 Report

In this manuscript authors propose  an inter-laboratory study of nanoparticle additivation effects on laser powder bed fusion of 2 metals and polymers. The information shown in the manuscript are new and the paper is well prepared. In my opinion authors could improve the title to highlight the theoretical character f the manuscript.  

Regards       

Author Response

Our responses to the Reviewer’s specific comments are as follows. Please note that the Reviewer’s comments are written in “red” color, our responses are written in “black”, the changes in the manuscript are marked in “blue” color, and new references are written in “light blue” color. Text changes are shown in the corresponding line number of the revised manuscript. In addition to responses, Fig. 6 and Fig. 9 are revised and references are reordered. Please see the revised manuscript in the attachment.

 Reviewer 1:

In this manuscript authors propose an inter-laboratory study of nanoparticle additivation effects on laser powder bed fusion of 2 metals and polymers. The information shown in the manuscript are new and the paper is well prepared. In my opinion authors could improve the title to highlight the theoretical character of the manuscript.

Thank you very much for the positive evaluation of our paper. We agree that the proposed ILS is not experimentally validated, making the ILS design more theoretical. Specifically to ILS, where multiple partners are involved and experimental redesign during the experimental cascade is almost impossible, the experimental design is a theoretical task of the highest importance. To that, our paper briefly reviews the powder, process, microstructure, and part property metrics affected by nanoparticle additivation in commercially available metal and polymer powder feedstocks and recommends the available characterization techniques to measure given property metrics. In addition, the importance of Research Data Management in such a wide-scale study is elaborated and supported with the proposed work and data flows. We, therefore, appreciate the reviewers’ idea of highlighting the theoretical character of the manuscript in the title and adapted it to the following:

“Nanoparticle additivation effects on laser powder bed fusion of metals and polymers – a theoretical concept for an inter-laboratory study design all along the process chain, including research data management

Reviewer 2 Report

This manuscript presents an interlaboratory study on the assessment of powder feedstock properties for the laser-based powder bed fusion process. The manuscript could be accepted for publication in Materials after considering the following MAJOR revisions: 

  1. The authors may mention the effort that other researchers have done to develop process-structure-property relationships in PBF-LB of several metals. The following references show examples of these efforts:
    • Debroy et al. (2018) Additive manufacturing of metallic components – Process, structure and properties
    • Yakout et al. (2017) The selection of process parameters in additive manufacturing for aerospace alloys
    • Yakout et al. (2018) A Review of Metal Additive Manufacturing Technologies
    • Dowling et al. (2020) A review of critical repeatability and reproducibility issues in powder bed fusion
  2. Although Figure 2 explains the workflow of the interlaboratory study for PBD-LB, the authors may describe data registration in their manuscript and the importance of data registration in part qualification. 
  3. It is also important to describe the design challenges such as CAD to STL conversion, data registration, CAD referencing, etc. and their influence on the workflow of ILS. 
  4. The authors should also emphasis on the influence of powder size distribution on the quality of the parts produced. There are several studies who identified this issue. 
  5. The novelty of this work compared to other ILS published work should be clarified in the conclusion section.

Author Response

Our responses to the Reviewer’s specific comments are as follows. Please note that the Reviewer’s comments are written in “red” color, our responses are written in “black”, the changes in the manuscript are marked in “blue” color, and new references are written in “light blue” color. Text changes are shown in the corresponding line number of the revised manuscript. In addition to responses, Fig. 6 and Fig. 9 are revised and references are reordered. Please see the revised manuscript in the attachment.

Reviewer 2:

This manuscript presents an interlaboratory study on the assessment of powder feedstock properties for the laser-based powder bed fusion process. The manuscript could be accepted for publication in Materials after considering the following MAJOR revisions:

We thank the Reviewer for the good evaluation of our work and his/her helpful suggestions to improve the quality of our manuscript.

The authors may mention the effort that other researchers have done to develop process-structure-property relationships in PBF-LB of several metals. The following references show examples of these efforts:

Debroy et al. (2018) Additive manufacturing of metallic components – Process, structure and properties

Yakout et al. (2017) The selection of process parameters in additive manufacturing for aerospace alloys

Yakout et al. (2018) A Review of Metal Additive Manufacturing Technologies

Dowling et al. (2020) A review of critical repeatability and reproducibility issues in powder bed fusion

We thank the Reviewer for listing valuable review papers focused on process-structure-property relationships in PBF-LB of metal powder feedstocks. We agree with the Reviewer that these references should be mentioned in our manuscript and have implemented them in several places in the new version in the following order:

[43] DebRoy, T.; Wei, H.L.; Zuback, J.S.; Mukherjee, T.; Elmer, J.W.; Milewski, J.O.; Beese, A.M.; Wilson-Heid, A.; De, A.; Zhang, W. Additive manufacturing of metallic components – Process, structure and properties, Progress in Materials Science 2018, 92, 112-224.

[44] Yakout, M.; Cadamuro, A.; Elbestawi, M.A.; Veldhuiset, S.C. The selection of process parameters in additive manufacturing for aerospace alloys, Int J. Adv Manuf. Technol. 2017, 92, 2081-2098.

[45] Dowling, L.; Kennedy, J.; O’Shaughnessy, S.; Trimble, D. A review of critical repeatability and reproducibility issues in powder bed fusion. Materials & Design 2020, 186, 108346, https://doi.org/10.1016/j.matdes.2019.108346

Line 59:

……. and enhancing part properties [13-24], and several reviews [2,3,43-45] are published to understand the process-structure-property relationships in the PBF-LB process.

We have further cited the reviews in the second part of our manuscript:

Line 347:

Powder bed density is directly linked to powder shape and powder size distribution during the PBF-LB process [45].

Line 600:

laser melting cannot be excluded. Furthermore, some metal alloy elements can be vaporized during laser melting that affects the solidification microstructure and part quality [43].

Please note that we have not cited Yakout et al. (2018) A Review of Metal Additive Manufacturing Technologies. This reference provides a review of key technologies for metal additive manufacturing, which does not meet the scope of our article.

Although Figure 2 explains the workflow of the interlaboratory study for PBD-LB, the authors may describe data registration in their manuscript and the importance of data registration in part qualification.

We agree with the Reviewer that data registration is of high importance. It is one of the most critical points of Research Data Management. However, we have not sensitized this in our initial version and implemented the following paragraph in the RDM section.

Line 696:

Here, data registration plays a critical role in the success of the ILS RDM. Each central laboratory will document the sampling techniques and test conditions while measuring each property metric, collect metadata, and generate datasets of related property metrics to improve data registration. All datasets with their corresponding documentation and SOPs will be stored in a central repository created by the coordination office. A long-term accessible repository will give open access to validated data for future data mining interests in the field of AM. Data reporting should be done according to ASTM F2971-13 [91]. In addition, ASTM is developing a new standard (ASTM WK73978 [92]) to comprise actions for data registration which will draw a route map for high-quality data registration under FAIR principles in the field of AM.

The following reference is cited in the revised manuscript:

[92] ASTM WK73978, New Specification for Additive Manufacturing - Data Registration, https://www.astm.org/DATABASE.CART/WORKITEMS/WK73978.htm, (Accessed on 12.08.2021).

It is also important to describe the design challenges such as CAD to STL conversion, data registration, CAD referencing, etc. and their influence on the workflow of ILS.

We agree with the Reviewer that CAD to .stl conversion can be challenging in very complex parts. Our paper describes an ILS design to test new material feedstocks, rather than creating complex 3D parts. For this purpose, printing is based on simple geometries (i.e., cubes, bars, and cylinders) where the CAD to .stl conversion is not challenging.

We would like to include this issue in the following paragraph to make it more straightforward for the readers:

Line 469:

…. before the start of ILS. CAD drawings will be converted to STL files by a central entity. Later, STL files will be distributed to each process participant to ensure that everyone uses the duplicate files. It is known that CAD to .stl conversion can be challenging in very complex parts. Our ILS design mainly addresses the powder material qualification and processability question rather than creating complex 3D parts. Hence, printing is based on simple geometries (i.e., cubes, bars, and cylinders) where the CAD to .stl conversion is not challenging. Each process participant will use their build processor for data preparation that fits their PBF-LB machine. 

The authors should also emphasis on the influence of powder size distribution on the quality of the parts produced. There are several studies who identified this issue.

The Reviewer is undoubtedly right about the influence of the powder size distribution on the quality of the parts. The PCA results in ref [2,3], for example, showed that powder size distribution/average powder size could be correlated with the Ultimate Tensile Strengths of as-built parts. Therefore, we included the following sentence (including new references) in the manuscript to show how part properties can be affected using different powder size fractions.

Line 347:

during the PBF-LB process [45]. Besides, different powder size fractions using the same process parameters influence the part quality [2,3, 71-73].

With the following new references:

[45] Dowling, L.; Kennedy, J.; O’Shaughnessy, S.; Trimble, D. A review of critical repeatability and reproducibility issues in powder bed fusion. Materials & Design 2020 186, 108346, https://doi.org/10.1016/j.matdes.2019.108346.

[71] Bonesso, M.; Rebesan, P.; Gennari, C.; Dima, R.; Pepato, A.; Calliari, I. Effect of Particle Size Distribution on Laser Powder Bed Fusion Manufacturability of Copper, Berg Huettenmaenn Monatsh 2021, 166, 256-262

[72] Brika, S.E.; Letenneur, M.; Dion, C.A.; Brailovski, V. Influence of particle morphology and size distribution on the powder flowability and laser powder bed fusion manufacturability of Ti-6Al-4V alloy, Additive Manufacturing 2020, 31, 100929, https://doi.org/10.1016/j.addma.2019.100929.

[73] P A Kuznetsov, I V Shakirov, A S Zukov, V V Bobyr’ and M V Starytsin, Effect of particle size distribution on the structure and mechanical properties in the process of laser powder bed fusion, Journal of Physics: Conference Series 2021, 1758, 1-9, doi:10.1088/1742-6596/1758/1/012021.

The novelty of this work compared to other ILS published work should be clarified in the conclusion section.

We thank the Reviewer for his/her valuable comment. We have extended the conclusion to summarize the differences to other studies:

Line 832:

………. along the process chain. In existing ILSs, a focus often lies on (e.g. part property) repeatability to manufacture specific parts (to see, e.g. machine-dependences) or to see inter-laboratory effects of general LPBF manufacturing of a material class (e.g. steel powder from varying sources). Such designs do not make sure that a process chain parameter can later be firmly attributed to another, not yet known one, via PCA. Also the strong focus on a) powder feedstock properties and b) RDM is unique. To provide a more concrete picture of the general theoretical concept and ILS design, and provide exemplary numbers to the proposed RDM concept, the presented exemplary design statistically evaluates the entire process chain by measuring 27 powder property metrics, 12 process property metrics, and 26 part property metrics. Of course, an extension or reduction of the number of varied material properties is easily possible based on the presented general ILS concept.  Further, the effect of nano-additives on the commercially available metal and polymer powder feedstocks and the repeatability of the modifications by NP can be measured for a better understanding of the PBF-LB process chain. Hence, this ILS design may be a blueprint to future ILS on LPBF, in particular in those ILS where powder feedstock effects shall be understood, and/or robust statistical analysis and PCA data processing is intended, where a good RDM design before ILS execution is key.       

Reviewer 3 Report

This study by the Ihsan Murat Kusoglu et al., reports on the inter-laboratory design to systematically study the effects of nanoparticle additivation on laser powder bed fusion (PBF-LB) of metals and polymers. The authors have discussed the in-detail list of analytical characterization methods to measure the powder feedstock and final part properties with a comprehensive research data management plan to achieve maximum efficiency in the PBF-LB process. This is a very good paper with the entire process chain and all important aspects of nanoparticle additivated laser powder bed fusion discussed in one place. These findings are likely to be an interest to the additive manufacturing community to develop materials with good properties, improve efficiency, and help better understand the structure-property-performance relationships. Overall, this is a well-written and interesting paper with motivation and contributions discussed by identifying and addressing a problem that has not been studied yet. Therefore, the reviewer did not find any issues with the paper and feels that it can be accepted by the journal in the present form.

Author Response

Our responses to the Reviewer’s specific comments are as follows. Please note that the Reviewer’s comments are written in “red” color, our responses are written in “black”, the changes in the manuscript are marked in “blue” color, and new references are written in “light blue” color. Text changes are shown in the corresponding line number of the revised manuscript. In addition to responses, Fig. 6 and Fig. 9 are revised and references are reordered. Please see the revised manuscript in the attachment.

Reviewer 3:

This study by the Ihsan Murat Kusoglu et al., reports on the inter-laboratory design to systematically study the effects of nanoparticle additivation on laser powder bed fusion (PBF-LB) of metals and polymers. The authors have discussed the in-detail list of analytical characterization methods to measure the powder feedstock and final part properties with a comprehensive research data management plan to achieve maximum efficiency in the PBF-LB process. This is a very good paper with the entire process chain and all important aspects of nanoparticle additivated laser powder bed fusion discussed in one place. These findings are likely to be an interest to the additive manufacturing community to develop materials with good properties, improve efficiency, and help better understand the structure-property-performance relationships. Overall, this is a well-written and interesting paper with motivation and contributions discussed by identifying and addressing a problem that has not been studied yet. Therefore, the Reviewer did not find any issues with the paper and feels that it can be accepted by the journal in the present form.

We thank the Reviewer for her/his constructive comments and interest in our paper. We believe that the proposed ILS design will be helpful in future research of the community as it leads to investigations of the entire process chain of AM, i.e., PBF-LB developing new powder feedstocks materials.

Reviewer 4 Report

With respect to materials-1321566: “Nanoparticle additivation effects on laser powder bed fusion of 2 metals and polymers – an inter-laboratory study design”

Although the proposed design of the ILS appears to be valid, and the paper provides clear background and is well organized, it is not a research paper in my opinion. You have not yet conducted any experimentation, so there is nothing here to be discussed. Your proposal could be useful and lead to significant research results in the future. Unfortunately, your work up to date is an untested proposal better suited to be published as a technical communication than as a research paper.

My recommendation is that you should re-evaluate which is the most adequate type of publication for this work.

Author Response

Our responses to the Reviewer’s specific comments are as follows. Please note that the Reviewer’s comments are written in “red” color, our responses are written in “black”, the changes in the manuscript are marked in “blue” color, and new references are written in “light blue” color. Text changes are shown in the corresponding line number of the revised manuscript. In addition to responses, Fig. 6 and Fig. 9 are revised and references are reordered. Please see the revised manuscript in the attachment.

Reviewer

Although the proposed design of the ILS appears to be valid, and the paper provides clear background and is well organized, it is not a research paper in my opinion. You have not yet conducted any experimentation, so there is nothing here to be discussed. Your proposal could be useful and lead to significant research results in the future. Unfortunately, your work up to date is an untested proposal better suited to be published as a technical communication than as a research paper.

My recommendation is that you should re-evaluate which is the most adequate type of publication for this work.

We thank the Reviewer for the critical comment and for highlighting the paper’s organization and transparent background.

We partly agree with the Reviewer’s comment; the different approach followed in our manuscript makes it differ from the most usual research papers published.

However, we believe that “research” is not limited to experimental papers, and feel supported by the positive, constructive assessment of our research paper by Reviewers #1-3. We feel that the paper will be quite helpful also for experimentalists among the Journal´s readers. The implementation of the proposed working procedure is not only intended within the inter-laboratory study. It is also proposed as a reference for other researchers working on the field of (PBF-LB) additive manufacturing, for the characterization, processing parameters, and data management protocols required to ensure the repeatability of the experiments in other laboratories (a must to ensure that the field advances according to the open science principles).

According to that, we do believe the manuscript contributes sufficient novelty to be published as a research paper. That is another reason why we intendedly submitted the manuscript to the Special Issue “Modeling, Simulation and Data Processing for Additive Manufacturing”, where the development of standards in terms of essential parameters reported, data management, and a clear process flow is a critical issue and a challenge in the field nowadays:

 https://www.mdpi.com/journal/materials/special_issues/additive_manufacturing_AM

Round 2

Reviewer 2 Report

The authors successfully responded to all reviewer’s comments; therefore, I recommend this manuscript for publication in Materials. 

Reviewer 4 Report

As I have argued in my previous review, I am concerned about the focus of the article, which seems more suitable as a technical communication than as a research article.

I believe that you have provided a valid proposal for the design of the ILS. However, a proposal is not "research", in my humble opinion, unless it provides a clear improvement that could be discussed either at a theoretical level or at an experimental level. This paper presents a theoretical proposal with no room for discussion, and nothing in your response motivates me to change my previous recommendation.